# Effects of Kettlebell Load on Joint Kinetics and Global Characteristics during Overhead Swings in Women

**DOI:** 10.3390/sports10120203

**Published:** 2022-12-11

**Authors:** Cullun Q. Watts, Kirsten Boessneck, Bryan L. Riemann

**Affiliations:** Biodynamics and Human Performance Center, Armstrong Campus, Georgia Southern University, Savannah, GA 31419, USA

**Keywords:** power, resistance training, kettlebell training, strength and conditioning, weightlifting

## Abstract

This study sought to identify the changes in ankle, knee, and hip joint kinetics with increasing load while performing the kettlebell overhead swing (OHS). Women (n = 18, age: 29.4 ± 5.3 years, 69.7 ± 8.9 kg) with a minimum of 6 months of kettlebell swing training experience performed fifteen repetitions of the kettlebell OHS with three different kettlebell masses (8 kg, 12 kg, 16 kg) in a counterbalanced order. Ankle, knee, and hip joint kinematics were captured within a 12-camera infrared motion capture space, while standing atop two force plates collecting ground reaction force (GRF) data. Post hoc results of statistically significant joint by mass interactions (*p* < 0.05) of the net joint moment impulse, work, and peak power revealed the hip demonstrating the greatest increase in response to load, followed by the ankle (*p* < 0.05). The knee joint kinetics changed very little between the masses. Pairwise post hoc comparisons between the joints at each mass level support the kettlebell OHS as being a hip dominant exercise, with the knee making the second largest contribution, despite contributions not changing across kettlebell masses. Collectively, these results provide practitioners with objective evidence regarding the mechanical demands and effects of load changes on the kettlebell OHS.

## 1. Introduction

Kettlebells are a non-traditional form of resistance training that utilizes a weighted ball with a handle attachment [1,2,3]. Their simplistic design allows for a wide array of unique movements as well as derivatives of common barbell movements such as the snatch [4] and clean [4]. Considering the small footprint and relatively low cost of kettlebells, it is unsurprising they have become a popular occupant of the strength and conditioning professional’s toolbox.

One of the most common kettlebell exercises are kettlebell swings, explosive movements featuring varying degrees of ankle, knee, and hip joint extension depending upon swing variation. Several training studies have examined the efficacy of kettlebell swing training on a variety of metrics, such as vertical jump [5,6,7,8], sprinting [5,9], and strength, including one repetition maximum for deadlifts [7], squats [6,8], and power cleans [8]. While some of the results have demonstrated kettlebell swing training to provide similar results as various resistance [5,6,7] or fitness [9] training programs, results within other studies have demonstrated resistance training to be superior to evoking performance enhancement [6,8]. Perhaps the different outcomes following kettlebell swing training interventions could be attributed to differences in the swing variations (e.g., overhead swings [OHS], shoulder height swings [SHS]), duration of training, training loads (e.g., kettlebell mass), or the progression of training loads through the training period. In order to optimize the evidence that practitioners have available for exercise prescription and progression [2,10], it is essential that objective documentation concerning the mechanical demands of an exercise, as well as how mechanical demands change with load progression [3,10], be available. When such objective documentation is available, practitioners can match client and patient goals with exercises that provide the needed training stimulus [10].

While simple qualitative observation yields kettlebell swings to largely involve the posterior muscle chain [6], particularly the hip extensors [11], current quantitative understanding across kettlebell swing variations and changes with progressive loads is limited. Several investigations have directly sought to quantify the mechanical demands of the various forms of kettlebell swings using either ground reaction force characteristics [2] or joint kinetics [1,3,12]; however, only two of these investigations considered the effects of kettlebell load progression on SHS [2,3]. Both reports relied upon pairwise post hoc comparisons to identify mechanical demand changes across three load progressions. Whereas such statistical analyses provide indication of whether subsequent sequential load increases are statistically significant or meet a certain effect size magnitude, they do not provide an indication regarding the pattern (e.g., linear, nonlinear) of mechanical demand changes across loads [10,13]. With regard to the investigations considering ankle, knee, and hip joint kinetics, one considered the effects of kettlebell mass on SHS between men and women [3], while a second compared joint kinetics between three swing styles (OHS, SHS, Indian club swings) in a mixed sex sample [1]. Interestingly the former investigation demonstrated several sex-related differences, suggesting future kettlebell swing research avoid mixed sample studies. Neither investigation statistically compared joint kinetic metrics between the ankle, knee, and hip joints. Thus, there is a void investigating the effects of kettlebell mass on ankle, knee and hip joint kinetics, as well as global mechanical characteristics, during kettlebell OHS in women. Given the apparent popularity of kettlebell OHS, particularly among more experienced kettlebell swingers and the CrossFit community, the primary purpose of the current investigation was to assess the effect of kettlebell mass on ankle, knee, and hip joint kinetics; namely the patterns of change in net joint moment impulse, work, and peak power during the concentric phase of a kettlebell OHS in young adult women. Consistent with a need to understand the global effects of progressive loads on mechanical demands, the secondary purpose was to determine the effect of kettlebell mass on global kettlebell swing performance and total system (kettlebell-participant sum) impulse, work, and peak power.

## 2. Materials and Methods

### 2.1. Subjects

Eighteen physically active women (29.4 ± 5.3 years, 69.7 ± 8.9 kg, 1.66 ± 0.05 m) with a minimum of 6 months of kettlebell training experience volunteered to participate in this study. Prior to any data collection, participants were given an overview of the study procedures, then signed an informed consent document before completing demographic and health history questionnaires related to musculoskeletal injuries and surgeries. All participants were in good health and were void of significant lower extremity or spinal injuries that prompted a restriction or change in physical activity participation within the past six months. Participants were asked to avoid vigorous physical activity 24 h prior to each session. Additionally, participants were excluded if they were unable to perform one set of 15 continuous swings with a 16 kg kettlebell. The OHS technique was derived from a previously documented form established by Bullock [1]. The university’s Institutional Review Board approved the study protocol. All subjects were informed of the risks and benefits of the testing and signed an informed consent document before testing.

### 2.2. Procedures

Each participant began the data collection session with a warmup consisting of 5 min on an arm ergometer (60–80 rpm), air squats, and 10 repetitions of 4.5–kg kettlebell swings. Next, reflective marker clusters were attached to the participant’s pelvis, as well as the participant’s dominant limb, thigh, shank, and foot. Fifteen swings were performed with each kettlebell (order randomly assigned) with two minutes rest between each set. Participants were not cued or trained on how to conduct the swings, including pace, except for being instructed to perform two-handed overhead swings according to how they typically trained. Prior to initiating each set of kettlebell swings, participants were required to stand upright and motionless for a minimum of 0.5 s; once 0.5 s time period expired, they were free to begin each continuous set of 15 swings.

### 2.3. Data Collection and Reduction

A 12 infrared camera system (Vicon, Oxford, UK) captured (100 Hz) three dimensional kinematic data of the participant marker clusters, as well as an additional marker cluster secured to each of the kettlebells. All camera data was streamed into The Motion Monitor software (IST, Chicago, IL, USA) where it was synchronized with ground reaction force data collected from two forceplates sampling at 1000 Hz (AMTI, Watertown, MA, USA), one under each foot. During participant setup and calibration, the proximal and distal ends of each body segment and the kettlebell mass center were digitized (centroid of contralateral points) using a marker cluster attached to a calibrated stylus. Additionally, the ankle, and knee joint centers were calculated by taking midpoints between contralateral points at each respective joint using the stylus. The hip joint center was established using a series of eight points along a circumduction cycle for each hip to estimate the apex of femoral motion. Subject’s mass and height were also recorded for anthropometric calculations required for locating each segment’s center of mass using the Dempster parameters as reported by Winter [14].

Ankle, knee, and hip joint angles and net joint moments were computed using The Motion Monitor after zero-phase lag low pass Butterworth filters were applied to the kinematic (10 Hz cutoff) and ground reaction force (35 Hz cutoff) data. These data, along with the vertical ground reaction force data, were exported as text files and further reduced using MatLab-based scripts (The Mathworks, Inc., Natick, MA, USA). All joint and kinematic data were visually examined and five consecutive swings in each set were selected for analysis. Criteria for selection included no missing marker data and continuous kettlebell movement (no pauses) with similar kettlebell displacements. Nearly all of the swings selected were between the 4th and 11th swings in the set of 15. The start, concentric-eccentric transition, and end of each selected swing was defined by the minima (start, end) and maxima (concentric-eccentric transition) of the combined vertical and anterior-posterior kettlebell displacement vector (square root of squared and summed vertical and anterior-posterior vectors). Joint angular velocities were calculated from the angular displacement time series data using the central finite difference method. Net joint moments were normalized to body mass, and for both the ankle and hip kinematic and kinetic data, the polarity of the data was reversed so that extension displacement and velocity and net joint extensor moments would be positive, thereby matching the knee. Net joint powers were computed as the product of the normalized net joint moments and angular velocity (radians/s) and net joint work was computed by integrating the net joint power-time data. Ankle, knee, and hip net joint moment impulses, work, and peak power were computed during the concentric phase only (start to concentric-eccentric transition).

Four kettlebell swing performance variables were computed: swing time, percent cycle, peak displacement, and peak velocity. Swing time was the time between the start and end of each selected repetition (includes both concentric and eccentric phases) and percent cycle was the percentage time at which the concentric-eccentric transition occurred. The difference between the start and concentric-eccentric magnitudes of the combined kettlebell vector, normalized to body height, determined peak displacement, while the peak kettlebell velocity, computed from the kettlebell displacement time series data using the central finite difference method, during the same time period was identified.

The sums of the anterior-posterior and vertical ground reaction forces across the two forceplates were computed. The system weight was computed based on the vertical ground reaction force sum for a minimum 0.3 s window based on visual inspection to verify quiet stance during the 0.5 s preceding the initiation of each kettlebell swing set. The net vertical ground reaction force was computed by subtracting the system weight from the vertical ground reaction forces. Vertical and horizontal acceleration of the total system were computed by dividing the net vertical and anterior-posterior ground reaction forces by system mass, respectively. Vertical and horizontal velocity of the total system was computed by integrating the acceleration data. Because the kettlebell swing involves both vertical and horizontal movement, the square root of squared and summed horizontal and vertical data were computed [2]. Net impulse, normalized to body mass, was computed by integrating the net total ground reaction force during the concentric phase of the swing. Power was computed as the product of net total ground reaction force and total system velocity. The maximum value, peak power, occurring during the concentric swing phase, normalized to body mass, was determined. Total system power was integrated during the concentric phase to compute the total system work (normalized to body mass).

### 2.4. Statistical Analysis

All statistical analyses were conducted using SPSS (version 27, IBM, Armonk, NY, USA). All outcome measures were averaged across the five selected swings and examined for normality using QQ plots and Shapiro–Wilk tests. Separate joint (ankle, knee, hip) by kettlebell mass (8 kg, 12 kg, 16 kg) repeated measures analysis of variance (RMANOVA) were conducted on the three joint kinetic variables (net joint moment impulse, work, and peak power). Post hoc analysis of significant interactions were conducted using trend analysis to reveal the effects of kettlebell mass on each of the three joints followed by Bonferroni adjusted simple main effect post hoc comparisons to identify differences between the three joints at each level of kettlebell mass. To determine the effects of kettlebell mass on the kettlebell swing performance variables (swing time, percent cycle, peak displacement, peak velocity) and total system variables (impulse, work, peak power) separate one way RMANOVA were conducted followed by post hoc trend analyses when appropriate. For all post hoc comparisons, standardized effect sizes were computed using Hedges’ g method, adjusted for small samples [15], and were interpreted as 0.2, 0.6, 1.2, 2.0 and 4.0 for small, moderate, large, very large, and extremely large, respectively [16]. Significance for all inferential statistics was set a priori to α < 0.05.

## 3. Results

### 3.1. Joint Kinetics

Kettlebell mass (Figure 1) had different effects on the ankle, knee, and hip net joint moment impulse (*p* = 0.001). While the ankle (*p* = 0.003, g = 0.79) and hip (*p* = 0.001, g = 0.91) demonstrated significant linear increases in net joint moment impulse, with the increase being significantly greater for the hip compared to the ankle (*p* = 0.016, g = 0.60), the knee demonstrated a significant quadratic trend (*p* = 0.019, g = 0.58). The ankle net joint moment impulse was significantly less than the knee (*p* = 0.009, g = 1.25) for the 8 kg kettlebell mass whereas there were no significant differences between the ankle and knee for the 12 kg (*p* = 0.167, g = 0.77) and 16 kg (*p* = 0.067, g = 0.95) kettlebell masses. Across all three kettlebell masses, the hip had significantly greater impulse than ankle (*p* < 0.001, g = 1.76 to 2.15) and knee (*p* < 0.001, g = 0.80 to 1.21).

Kettlebell mass (Figure 2) also had different effects on ankle, knee, and hip work (*p* < 0.001). While the ankle (*p* = 0.002, g = 0.81) and hip (*p* < 0.001, g = 1.01) demonstrated significant linear increases in work, with the increase being significantly greater for the hip compared to the ankle (*p* < 0.001, g = 0.92), the knee demonstrated a significant quadratic trend (*p* = 0.010, g = 0.65). For each kettlebell mass, knee work was significantly greater than the ankle (*p* < 0.002, g = 1.37 to 1.70) and hip work was significantly greater than the knee (*p* < 0.001, g = 1.48 to 1.85).

Kettlebell mass (Figure 3) also had different effects on ankle, knee, and hip peak power (*p* < 0.001). While the ankle (*p* = 0.012, g = 0.63) and hip (*p* = 0.003, g = 1.11) demonstrated significant linear increases in work, with the increase being significantly greater for the hip compared to the ankle (*p* = 0.005, g = 0.73), while the knee demonstrated a significant quadratic trend (*p* = 0.009, g = 0.66). For each kettlebell mass, knee work was significantly greater than the ankle (*p* < 0.001, g = 1.43 to 1.85) and hip work was significantly greater than the knee (*p* < 0.001, g = 1.13 to 1.45).

### 3.2. Kettlebell Swing Variables

Kettlebell mass had significant effects on swing time (*p* < 0.001), peak kettlebell displacement (*p* < 0.001), and peak kettlebell velocity (*p* < 0.001) (Table 1). Swing time (*p* < 0.001, g = 1.05) and peak kettlebell displacement (*p* < 0.001, g = 0.99) increased linearly with increasing kettlebell mass, whereas peak kettlebell velocity (*p* < 0.001, g = −3.23) decreased linearly with increasing ball mass. The quadratic trends for swing time (*p* = 0.485, g = 0.16), peak kettlebell displacement (*p* = 0.092, g = 0.40), and peak kettlebell velocity (*p* = 0.426, g = 0.18) were not statistically significant. Kettlebell mass had no significant effect on percent cycle (*p* = 0.070).

### 3.3. Total System Variables

Kettlebell mass had significant effects on total system impulse (*p* < 0.001), work (*p* < 0.001), and power (*p* < 0.001) (Table 2). Total system impulse (*p* < 0.001, g = 1.81), work (*p* < 0.001, g = 2.67), and power (*p* < 0.001, g = 1.49) demonstrated significant linear increases across kettlebell masses. Additionally, whereas impulse exhibited a significant quadratic increase across kettlebell masses (*p* = 0.048, g = 0.50), the quadratic trends for work (*p* = 0.101, g = 0.41) and power (*p* = 0.638, g = 0.11) were not statistically significant.

## 4. Discussion

With the aim of providing practitioners with evidence quantifying the biomechanical demands of kettlebell OHS exercise, the specific focus of the current investigation was to determine the effects of kettlebell mass on concentric ankle, knee, and hip joint kinetics in women. Secondarily, we sought to determine the effect of kettlebell mass on global markers of kettlebell performance and total system impulse, work, and peak power. In contrast to the knee, both the ankle and hip joint kinetics increased across the kettlebell masses with the changes within each joint being similar between subsequent loads; however, the increases for the hip were significantly greater. The joint kinetic results clearly support the kettlebell OHS as being a hip dominant exercise and the knee making the second largest contribution. With the exception of percent cycle and total system impulse, the kettlebell swing performance and total system results revealed equal changes between each subsequent load increase. Total system impulse uniquely demonstrated a larger increase between the 12 kg and 16 kg kettlebell masses than between the 8 kg and 12 kg kettlebell masses. Collectively, these results provide practitioners with objective evidence regarding the mechanical demands and effects of load changes on kettlebell OHS.

Intuitively there is an expectation that the external loads used for an exercise directly influences the mechanical demands of the exercise. In contrast to conducting multiple post hoc pairwise and complex comparisons of significant load effects, our statistical approach used post hoc trend analyses. In addition to avoiding the reduction in statistical power associated with adjusted comparison alpha levels, using post hoc trend analyses provides a more simplistic approach to determining the overall pattern of changes across an ordinal independent variable (i.e., load), as well as facilitating the comparison of the changes in magnitude between the levels of a second nominal independent variable (i.e., ankle, knee, hip). Significant linear trends in the current investigation indicates equal changes in the metric between load levels; whereas, significant quadratic trends indicate unequal changes between load levels.

Considering joint kinetics provides insight into the mechanical demands at each joint for a given exercise [10,13,17]. To obtain a comprehensive perspective of the training stimulus to the ankle, knee and hip joint extensors, the current investigation considered three perspectives: net joint moment impulse, work, and power. Impulse and work provide insight regarding net torque production over time and torque production through the range of motion, respectively. Joint power, the product of torque and angular velocity, reflects the rate at which mechanical energy is being produced. Regardless of the different perspective each joint kinetic metric provides, the effects of kettlebell mass were largely consistent between net joint moment impulse, work, and power. Based upon the ankle and hip eliciting significant linear trends across kettlebell masses, the mechanical demands within each joint changed similarly between each incremental 4 kg kettlebell mass increase; direct comparison of the trends revealed the hip increases to be significantly greater (moderate effect size) than the ankle. Thus, regardless of the joint kinetic metric, increasing kettlebell mass will prompt an increased mechanical demand from the hip extensors and secondarily the ankle plantar flexors. The hip exhibiting the largest joint kinetic changes with increasing loads, followed by the ankle, is consistent with a previous report examining peak net joint moments during SHS under three load conditions [3].

Surprisingly, rather than exhibiting linear increases across kettlebell loads, the knee demonstrated significant quadratic trends (moderate effect size) across kettlebell loads for all three joint kinetic metrics. Close inspection of the pattern across the three loads reveals a slight decrease for the 12 kg kettlebell mass compared to the 8 kg and 16 kg masses. The decrease appears to be of sufficient magnitude to prompt the quadratic trend reaching statistical significance. While we cannot explain with certainty, there may have been a slight shift in technique that occurred between the 8 kg and 12 kg kettlebell masses that prompted a decrease in knee joint kinetics. The lack of a significant linear trend indicates the knee joint did not contribute to the increased demands imposed as kettlebell mass increased. Interestingly, while the kettlebell OHS technique is described as being more similar to a squatting motion compared the SHS being considered more similar to a deadlift movement [18], Levine et al. [3] reported small increases in knee peak net joint moments across three kettlebell masses during SHS.

Consistent for the three joint kinetic metrics considered, barring the exceptions of no significant differences between the ankle and knee for net joint moment impulse with the 12 kg and 16 kg kettlebells, the current results clearly demonstrate the kettlebell OHS as prompting the greatest contribution from the hip, followed by the knee and then the ankle. The magnitudes of the effect sizes between the joints ranged from large to very large. In contrast, although formal statistical comparisons were not conducted between the ankle, knee and hip joints, the peak net joint moment descriptive statistics provided by Bullock et al. [1] for kettlebell OHS indicate the ankle was higher than the knee. It is worthwhile to note that the standard deviations reported for the peak knee net joint moments were 65% of the mean magnitudes reflecting considerable between-participant variability. The peak joint powers reported by Bullock et al. [1] are consistent with our results of the hip being the highest, followed by the knee and then the ankle.

Although the three kettlebell masses in the current kettlebell OHS study were half (8 kg, 12 kg, 16 kg) the masses used by Lake and Lauder [2] (16 kg, 24 kg, 32 kg) with SHS, comparable total system results (impulse and peak power) were revealed. The rationale for using lighter kettlebell masses in the current study was related to using women participants. Lake and Launder [2] reported significant impulse changes with each increase in kettlebell mass. Similarly, impulse in the current study increased with load (large effect linear trend), but the change between the two heavier kettlebell masses was significantly greater than the two lighter kettlebell masses (moderate effect quadratic trend). In the current investigation, peak power demonstrated similar increases between kettlebell loads (large effect linear trend), whereas Lake and Lauder [2] reported no significant difference in peak power between the two lighter kettlebell masses. In both studies, peak power continued to increase between the two heaviest kettlebell loads. Future studies should incorporate heavier mass kettlebells to identify the kettlebell load associated with peak power production. Interestingly, despite different kettlebell masses used in the two studies, with impulse normalized to body mass, very similar magnitudes resulted. In contrast, the peak power magnitudes reported by Lake and Lauder [2] were double the magnitudes in the current study. Perhaps the peak power differences are attributable to swing variations (i.e., OHS versus SHS). Finally, based upon the kettlebell swing trajectory incorporating both anterior and vertical components during the swing, both investigations computed total system metrics by combining the vertical and anterior-posterior ground reactions forces. This point is relevant for practitioners to consider with regard to specificity of training and the mechanical demands of some sport movements [2].

Of the three kettlebell swing characteristics that demonstrated significant kettlebell load effects, based upon the linear trend effect size magnitudes, peak concentric velocity was impacted the greatest (extremely large effect size). As would be expected and consistent with a previous report [2], each subsequent increase in kettlebell mass resulted in a peak concentric velocity reduction. Interestingly, the peak concentric velocities reported by Lake and Lauder [2] during SHS using 16 kg, 24 kg and 32 kg mass kettlebells were within 0.25 m/s of the peak velocities recorded in the current study. The linear trend effect size magnitudes for swing time and peak displacement were near identical (large effect sizes) suggesting that kettlebell mass influenced these variables similarly. With regard to swing time, because the percent of the swing cycle at which concentric-eccentric phase transition occurred was not significantly different between kettlebell masses, we are unable to attribute the increases in swing time to either the concentric or eccentric phases. Across all three kettlebell masses, based upon the percent cycle values ranging between 48.3% to 50.2%, a near equal quantity of time was spent in the concentric and eccentric phases of the swings. The swing time for the 16 kg kettlebell mass in the current study was extremely close (~0.01 s) to a previous kettlebell OHS study [1]; however, it is important to note that the previous study reported an overall swing time average from a mixed sex sample with the women using 12 kg and men using 20 kg kettlebells. In the current study, peak displacement increased approximately 2% BH from the 8 kg to 16 kg kettlebells which corresponds to an approximately 3.3 cm greater displacement. We attribute the increase to greater displacement (more posterior and vertical) occurring at the end of the eccentric phase. Interestingly, Lake and Lauder [2] reported no significant changes in SHS kettlebell displacement across the three loads considered.

Our study has a few limitations. First, participants were required to have at least 6 months of OHS training. The exact OHS training experience and resistance training age of the participants was not controlled. Thus, there is a possibility that joint kinetic patterns change with OHS training or resistance training experience. Future research should examine this possibility. Additionally, the participants had learned to perform OHS prior to study participation and there was no effort during study procedures to provide any technique cues or feedback other than the requirement to complete the swings with the kettlebell moving overhead. As a result, there may have technique variations between participants which responded differently to the changing kettlebell loads. Furthermore, data collection was conducted during a single session. Participants were given time to practice several swings after the marker clusters were secured to body segments, but it is unknown if the practice was sufficient for participants to become completely accustomed and consistent in performing the swings with the clusters. Finally, the highest load of kettlebells used was 16 kg which may have limited the ability of participants to maximally express power. Future research should consider joint kinetics associated with kettlebells having greater mass.

## 5. Conclusions

This study is the first to quantify the effects of load on the joint kinetics of the kettlebell OHS. As the kettlebell mass increased, the greatest increases in joint kinetics occurred primarily at the hip, followed by the ankle. Contrary to the previous kettlebell SHS investigation, the current results did not demonstrate changes for the knee across the loads. Total system peak power, work and impulse were found to be highest while performing the kettlebell OHS with the heaviest load condition, which is in agreement with the previous SHS investigation. This study confirms that the kettlebell OHS is a hip-dominant movement similar to the kettlebell SHS. Practitioners should consider the kettlebell OHS as an option when they are looking to develop lower body power output or specifically hip concentric power without concern for, or separately from the knee. Future research should directly compare the joint kinetics of the kettlebell OHS and the kettlebell SHS while under several kettlebell loads to investigate the possibility of a direct relationship between the load used for a kettlebell SHS and the load required by a kettlebell OHS to produce similar joint kinetics.

## Figures and Tables

**Figure 1 sports-10-00203-f001:**
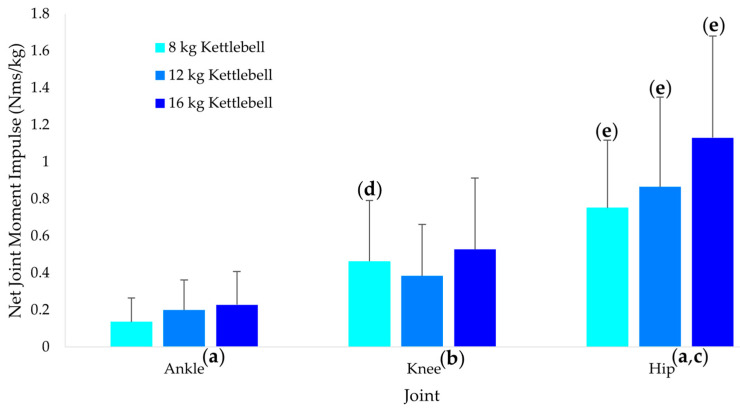
Ankle, knee and hip net joint moment impulses across the three kettlebell masses. (**a**) Significant linear trend (*p* < 0.05). (**b**) Significant quadratic trend (*p* < 0.05). (**c**) Linear trend greater than at the ankle (*p* < 0.05). (**d**) Greater than at the ankle for the same weight condition (*p* < 0.05). (**e**) Greater than at the ankle and knee for the same weight condition (*p* < 0.05). The values and error bars represent the means and standard deviations respectively.

**Figure 2 sports-10-00203-f002:**
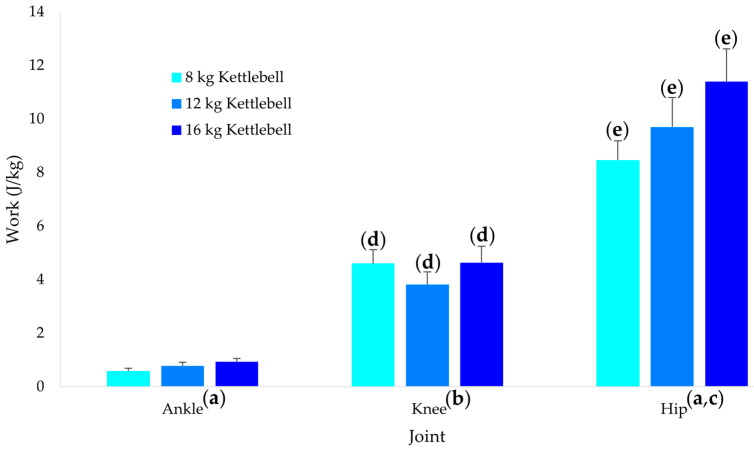
Ankle, knee and hip work across the three kettlebell masses. (**a**) Significant linear trend (*p* < 0.05). (**b**) Significant quadratic trend (*p* < 0.05). (**c**) Linear trend greater than at the ankle (*p* < 0.05). (**d**) Greater than at the ankle for the same weight condition (*p* < 0.05). (**e**) Greater than at the ankle and knee for the same weight condition (*p* < 0.05). The values and error bars represent the means and standard deviations respectively.

**Figure 3 sports-10-00203-f003:**
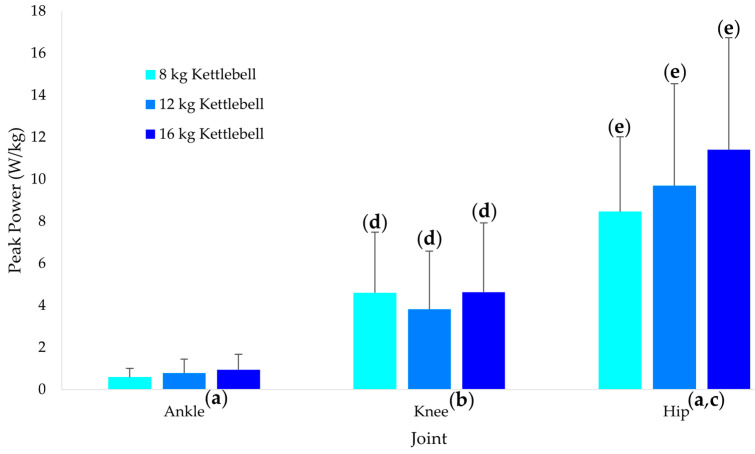
Ankle, knee and hip peak power across the three kettlebell masses. (**a**) Significant linear trend (*p* < 0.05). (**b**) Significant quadratic trend (*p* < 0.05). (**c**) Linear trend greater than at the ankle (*p* < 0.05). (**d**) Greater than at the ankle for the same weight condition (*p* < 0.05). (**e**) Greater than at the ankle and knee for the same weight condition (*p* < 0.05). The values and error bars represent the means and standard deviations respectively.

**Table 1 sports-10-00203-t001:** Kettlebell swing performance results.

	8 kg	12 kg	16 kg
Swing time (s) *	1.82 ± 0.18	1.88 ± 0.18	1.91 ± 0.17
Percent cycle (%)	48.34 ± 5.1	48.87 ± 4.4	50.19 ± 4.0
Peak displacement (%BH) *	102.0 ± 4.5	103.5 ± 4.6	104.1 ± 4.1
Peak velocity (m/s) *	4.78 ± 0.25	4.57 ± 0.25	4.34 ± 0.25

BH: body height; *: Significant linear effect of kettlebell mass; The values represent the means and standard deviations respectively.

**Table 2 sports-10-00203-t002:** Total system results.

	8 kg	12 kg	16 kg
Impulse (Ns/kg) * †	2.40 ± 0.48	2.59 ± 0.46	3.08 ± 0.54
Work (J/kg) *	4.34 ± 0.85	5.32 ± 0.82	6.95 ± 1.12
Peak Power (W/kg) *	12.16 ± 2.23	13.72 ± 3.49	15.84 ± 3.41

BH: body height; *: significant linear effect of kettlebell mass; †: significant quadratic effect of kettlebell mass; The values represent the means and standard deviations respectively.

## Data Availability

The data presented in this study are available on request from the corresponding author. The data are not publicly available due to ongoing analysis for future study planning.

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
