# Peer review of "Effects of Kettlebell Load on Joint Kinetics and Global Characteristics during Overhead Swings in Women"

_sports, 2022, doi:10.3390/sports10120203_

Round 1
Reviewer 1 Report
This study clarified the effect of loads on the joint kinetics during the kettlebell exercise. The main finding was that as the kettlebell mass increased, the greatest increases in the joint kinetics occurred primarily at the hip, followed by the ankle. This paper is well written by good English. The results are clearly presented and the conclusions are hardly controversial. The main research findings of this paper will be important for the full understanding of joint kinetics during the kettlebell exercise. I believe that the data are potentially interesting for the practitioner and coach.
2. Materials and Methods
Lines 105-107: Clarify the sampling frequency of the force plates.
3. Results
Add the typical example of time-series data of joint moment, joint angular velocity and power using the figures.
Figure 3. Change the name of vertical axis to peak power.
Table 1. The mean value of 8 kg is 102.0? Check all significant digits.
4. Discussion
Lines232: increases for the hip were significantly greater for the hip??
Add limitation in this study.
Reviewer 2 Report
It would be helpful to describe the technique of the OHS. I would also suggest including photos of the technique. There are limitations that should be stated such as the reliability of the measures, and only one testing session. However, about testing conditions e.g., time of day, whether participants were given any special instructions such as what to avoid prior to testing. It seems possible that participants could have changed techniques/performance due to other factors besides only the load of the kettlebell. Please discuss this briefly. I do not recall whether the sequence of loads used for the OHS was randomized. Below are some further comments.
Line 35: The following part of this sentence is difficult to understand “….to have equivalent effects as various resistance…”. Could you please revise it?
Line 39: “..attributable to differences in the swing variations…”
Lines 85-86: Did you use a criterion to assess the ability to perform kettlebell swings?
Line 93: “…10 repetitions of 4.5-kg kettlebell swings.”
Line 94: “…clusters were attached…”
Line 100: 0.5 seconds? Is this correct?
Line 147: “…preceding the initiation…”
Line 185: “…no significant differences between…”
Lines 179-188: If the effect sizes are based on Hedges’ g, then effect sizes should be g = XX and not d = XX.
Line 185: An effect size of 0.77 and p-value of 0.167. Is this correct?
Figure 1 and Figure 3: Spell out ‘NJMI’
Line 233: “…the increases for the hip were significantly greater for the hip.” – This is a little confusing. Please revise.
Lines 253-254: This sentence is incomplete.
Line 275: “…appears to be of sufficient magnitude….”
Line 277: ‘….between that occurred between the 8kg and 12kg…’ – Please revise.
Lines 284-287: Break up into two concise sentences.
Round 2
Reviewer 2 Report
Well done on addressing my concerns.
Author Response
We appreciate the reviewers efforts in improving our manuscript.